# MCM: Masked Cell Modeling for Anomaly Detection in Tabular Data

**Jiaxin Yin**[†*], **Yuanyuan Qiao**[†*], **Zitang Zhou**[†], **Xiangchao Wang**[‡], **Jie Yang**[†]
[†]Beijing University of Posts and Telecommunications, Beijing, China, [‡]Hangzhou Dianzi University
`{yinjx, yyqiao, zhouzitang, janeyang}@bupt.edu.cn,20041926@hdu.edu.cn`

## Abstract

This paper addresses the problem of anomaly detection in tabular data, which is usually implemented in an one-class classification setting where the training set only contains normal samples. Inspired by the success of masked image/language modeling in vision and natural language domains, we extend masked modeling methods to address this problem by capturing intrinsic correlations between features in training set. Thus, a sample deviate from such correlations is related to a high possibility of anomaly. To obtain multiple and diverse correlations, we propose a novel masking strategy which generates multiple masks by learning, and design a diversity loss to reduce the similarity of different masks. Extensive experiments show our method achieves state-of-the-art performance. We also discuss the interpretability from the perspective of each individual feature and correlations between features. Code is released at `https://github.com/JXYin24/MCM`.

## 1 Introduction

Tabular data is the most widespread data type in anomaly detection (AD), which is crucial for various scientific and industrial processes, such as medical disease detection (Fernando et al., 2021), financial fraud detection (Al-Hashedi & Magalingam, 2021), network intrusion detection (Ahmad et al., 2021), etc. In most real-world applications, it's usually impractical or prohibitive to obtain labeled anomalies. Therefore, AD in tabular data is usually implemented in an one-class classification setting where only normal samples are accessible in training set. The key of AD under this setting is to extract characteristic patterns of training data, such that anomalies can be detected due to a deviation from these patterns (Ruff et al., 2021). However, because the features of tabular data are heterogeneous, complex, and have no fixed structure (Chang et al., 2023), finding such "characteristic patterns" is quite challenging.

Self-supervised learning (SSL) methods can address this challenge by creating pretext tasks to train neural networks (NN) to learn characteristic patterns within training data. In particular, GOAD (Bergman & Hoshen, 2020) uses a distance based pretext task, and leverages a NN to map data to a latent space, in which the distances of a training sample to the cluster center after applying each transformation are minimized. The characteristic patterns are modeled by such distances in GOAD. NeuTral AD (Qiu et al., 2021) and ICL (Shenkar & Wolf, 2022) both employ contrastive learning based loss functions to create pretext tasks. NeuTral AD first designs a set of learnable transformations, then pulls the representation of each transformed sample close to the original sample, and pushes representations of different transformed samples away from each other. ICL divides a sample into several pairs, each pair contains a subset of consecutive features and a subset of the rest features. Then, ICL maximizes the mutual information between subsets from the same pairs and minimizes that from different pairs. Characteristic patterns in above two methods are just modeled by their contrastive losses, so a sample with a high loss value indicates a high possibility of anomaly. However, no work has adapted masked image/language modeling (MIM/MLM), the most popular and effective task in SSL (Schiappa et al., 2023), to address the problem of tabular AD.

In this paper, we mitigate this gap by MCM: masked cell modeling for AD in tabular data. Inspired by MIM (Bao et al., 2021; He et al., 2022) and MLM (Devlin et al., 2018; Radford et al., 2018) which

---

[*]Corresponding author

reconstruct masked patches and tokens by capturing their correlations between unmasked patches for images and contexts for texts, we train MCM to capture intrinsic correlations between features existing in training tabular data, and model the "characteristic patterns" by such correlations. For masked modeling tasks, we try to capture the correlations by finding optimal masks, under which the masked features of training data can be restored well with access to only unmasked features. In this way, data deviating from a correlation can be detected by a large reconstruction error under corresponding mask. Since finding such "masks" manually is quite challenging, we propose a learnable masking strategy and leverage NN to find such "masks".

Concretely, we design a mask generator, which takes original data as input and outputs multiple masking matrices. The masking matrices will perform element-wise product with the input to produce multiple masked data. Due to the constraint of reconstruction loss, the network will search for masking matrices under which the masked features can be restored well with unmasked features. Furthermore, we design a diversity loss, which constrains the similarity of different masking matrices, preventing generating redundant masks and rendering different masks to capture diverse correlations existing in normal data. Overall, the main contributions can be summarized as follows:

- We propose MCM, a masked cell modeling framework for AD in tabular data. We design a learnable masking strategy for MCM, which can determine how to mask for each input through training. To our best known, it's the first time to extend the success of masked image/language modeling to the domain of tabular AD.
- We leverage ensemble learning and design a diversity loss to make MCM capture multiple and diverse correlations in training data, providing more information to characterize normal data and distinguish anomalies. We also provide interpretability of MCM from two perspectives: correlations between features and abnormality of each independent feature.
- Experimental results on various AD datasets demonstrate that MCM achieves state-of-the-art performance. We also investigate the effect of different masking strategies for MCM and provide an extensive experimental analysis with ablation studies.

## 2 RELATED WORK

**Classic AD Methods** A number of classic methods are proposed for AD over past several decades. Probability based methods first leverage a parametric or nonparametric distribution to fit the normal data, then detect anomalies based on the probability that the data appears in this distribution. Representative works are using kernel based density estimator (Pavlidou & Zioutas, 2014), gaussian mixture models (Yang et al., 2009), and empirical cumulative distribution (Li et al., 2022). Distance based methods evaluate test points with respect to their distance with other instances. For example, KNN (Ramaswamy et al., 2000) calculates the average euclidean distance of a test point with its nearest points as anomaly score while LOF (Breunig et al., 2000) uses the ratio of local reachability density of a sample and its nearest neighbors. Classification based methods aim to directly learn a decision boundary with access to data only from normal class. OCSVM (Schölkopf et al., 2001) generates this boundary by achieving the maximum separation between the input data and the coordinate origin. Tax & Duin (2004) try to learn a smallest hypersphere that can surround most samples, so data out of the hypersphere can be judges as anomalies. Unfortunately, classic methods suffer from curse of dimensionality and a relatively low detection accuracy.

**Self-Supervised Learning for AD** Several recent studies have revealed that SSL methods can address the issue of curse of dimensionality and improve detection accuracy in AD. For instance, Golan & El-Yaniv (2018) and Wang et al. (2019) train a classifier to discriminate different geometric transformations applied on given images, at test time, softmax activations predicted by the classifier on transformed images are utilized for detecting anomalies. Bergman & Hoshen (2020) extend the applicability of transformation-based methods to tabular data by changing geometric transformations to random affine transformations. Some works (Tack et al., 2020; Sehwag et al., 2021; Sohn et al., 2021) try to first learn representations by a contrastive learning objective (Chen et al., 2020) and then judge anomalies based on the representations. Differently, Tack et al. (2020) introduce an additional classifier when training and combine the information from representations and the classifier for prediction. Sehwag et al. (2021) use Mahalanobis distance based detection in the representation space while Sohn et al. (2021) directly apply an one-class classifier on the representations. Further efforts focus on generalizing contrastive learning to AD in tabular data. Qiu et al. (2021) achieve

this by designing learnable transformations and a novel deterministic contrastive loss, which pulls the representation of each transformed sample close to the original sample, and pushes them away from different transformed samples. Shenkar & Wolf (2022) employ an internal contrastive loss, consider one sample at a time and match a subset of its features with the rest. However, no work has extended the success of MIM/MLM to tabular AD. We propose MCM to mitigate the gap.

**Masked Image/Language Modeling** The advent of BERT (Devlin et al., 2018) and GPT (Radford et al., 2018; 2019; Brown et al., 2020) significantly advance performance across various NLP tasks. These methods utilize a MLM objective where a portion of the tokens of input sequence are held out and models are trained to learn representations by predicting the masked tokens. BEiT (Bao et al., 2021) extends the success of MLM to CV domain, which first tokenizes the original image into discrete tokens and then learns representations by BERT-style pretraining. Based on this, PeCo (Dong et al., 2023) adopts a perceptual loss, enforcing the tokens to have rich perceptual information. MAGE (Li et al., 2023) uses variable masking ratios to unify image generation and representation learning. Another line of MIM approaches try to predict masked pixels instead of discrete tokens, the most representative work is MAE (He et al., 2022). SimMIM (Xie et al., 2022) simplifies the decoder with one-layer prediction head while CAE (Chen et al., 2023) employs a latent regressor to decouple the role of representation learning and pretext task completion. However, above methods heavily rely on the intrinsic structure of language and image data, and cannot generalize to tabular data directly. In this paper, we propose a learnable masking strategy and design a loss function suitable for tabular data, extending the success of MIM/MLM to tabular AD.

# 3 METHOD

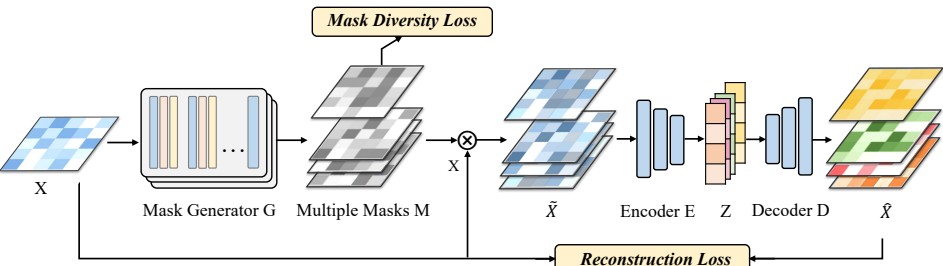

Figure 1: The proposed model. MCM consists of three modules, a mask generator, an encoder and a decoder. The mask generator generates multiple mask matrices, which will perform element-wise product with input data to obtain their masked versions. The encoder encodes the masked data to latent representations, and the decoder reconstructs original data from the representations.

## 3.1 OVERVIEW

We show the overview of our proposed MCM in Fig.1. Given a set of input samples $X$, the mask generator takes them as input, and outputs multiple mask matrices. Then, $X$ will perform element-wise product with every mask matrix to produce multiple masked inputs. Afterwards, an encoder maps the masked inputs to latent representations while a decoder then remaps the representations from latent space to original space for reconstruction. The whole framework is trained end-to-end under the constraint of a reconstruction loss and a diversity loss. After training, average reconstruction error of a test sample with it's all reconstructed versions is used as anomaly score.

## 3.2 MASKING STRATEGY

To capture intrinsic correlations existing in training data, we propose a learnable masking strategy, producing masks under which the masked features can be restored well with access to only unmasked features by learning.

Specifically, We employ a mask generator $G$, which consists of a feature extractor $F$ attached by a sigmoid function. $G$ learns information from the input data $X$, and outputs a masking matrix $M$ with

the same dimension with $X$. Because of the sigmoid function, each value of $M$ is ranging from 0-1. Compared to regular binary mask with a mask value of either 0 or 1, this method provides a more flexible degree of masking. Each row of $M$ is the mask for one sample $\mathbf{x}$ across its different features, and each column is the mask for one feature across different training data. It can be seen this masking strategy is not only learnable, but also data-related, i.e. assigning different masks for different data, which contributes to finding salient correlation for a specific normal sample. After training, when inputs a test sample, MCM will first produce a mask that models a correlation training data should match. Because all the training data are normal samples, this correlation tends to be specific for normal data, anomalies are prone to deviate from this correlation and thus be detected by a larger reconstruction error.

### 3.3  ENSEMBLE AND DIVERSITY

Considering correlation modeled by just one mask is not enough to distinguish normal data from others, we propose to generate multiple masks to capture multiple correlations in training data. We adopt ensemble learning method, assemble multiple feature extractors $F_1, F_2, ..., F_K$ to the mask generator and make it produce multiple mask matrices $M_1, M_2, ..., M_K$.

$$M_1, M_2, ..., M_K = G(X) = sigmoid(F_1(X), F_2(X), ..., F_K(X)) \tag{1}$$

Then, multiple masked inputs $\tilde{X}_1, \tilde{X}_1, ..., \tilde{X}_K$ are obtained by the element-wise product of $X$ and $M_1, M_2, ..., M_K$, which will be sent to the encoder $E$ and the decoder $D$ to obtain reconstructed data $\hat{X}_1, \hat{X}_1, ..., \hat{X}_K$:

$$\tilde{X}_k = X \odot M_k, \quad \hat{X}_k = D(E(\tilde{X}_k)), \quad k = 1, 2, ..., K. \tag{2}$$

Through reconstruction task, multiple correlations in training data are captured by multiple masking matrices.

Still, it raises another vital problem: how to prevent the mask generator generating same and redundant masks. This determines whether MCM can extract different and diverse correlations. Our solution is to constrain the similarity between different masking matrices. Because these matrices are learnt by neural networks, their values can be guided by loss function through back-propagation. We calculate the sum of the similarities between all the matrices and add this term to loss function, successfully improving the diversity of different masks.

We visualize the reconstruction error of several samples under 15 different masks in Fig.2. As is shown in this figure, it's possible some anomalies conform to some kind of correlation captured in normal data, such as the correlation built by 10 mask for #5 anomaly (reflected in a low reconstruction error). But when capturing multiple and diverse characteristic correlations, it's almost impossible for an anomaly to match all the correlations existing in normal data (reflected by a larger reconstruction error on most masks for anomalies). Thus, the ensemble of diverse correlations can improve the detection ability of MCM.

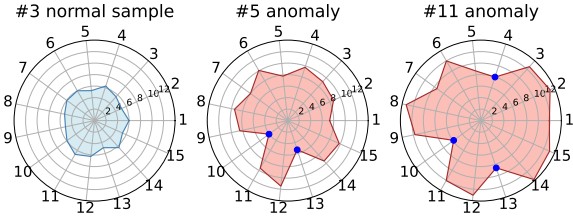

Figure 2: Radar chart of one normal and two abnormal samples of Breastw dataset, the points denote reconstruction error under different masks.

### 3.4  LOSS FUNCTION

We present the following overall loss to train our model:

$$\mathcal{L} = \mathcal{L}_{rec} + \lambda \mathcal{L}_{div} \tag{3}$$

where $\mathcal{L}_{rec}$, $\mathcal{L}_{div}$ are reconstruction loss and diversity loss, respectively, $\lambda$ is a predefined hyperparameter to balance them.

**Reconstruction Loss** is extensively used in mask modeling tasks. In our setting, it minimizes the $l_2$-norm of all the reconstructed data from different masked versions with their corresponding unmasked version. Averaging operation is used twice, the first is for different masks and the second is for different samples:

$$\mathcal{L}_{rec} = \frac{1}{K} \sum_{k=1}^{K} \|\hat{X}_k - X\|_F^2 = \frac{1}{NK} \sum_{i=1}^{N} \sum_{k=1}^{K} \|\hat{\mathbf{x}}_k^{(i)} - \mathbf{x}^{(i)}\|_2^2 \qquad (4)$$

where superscript $i$ denotes the index of samples and subscript $k$ denotes the index of different masks. N, K represents the total number of training data and masks, respectively.

**Diversity Loss** is to encourage different masks to focus on diverse correlations existing in normal data. We achieve this purpose by constraining the similarity of different masking matrices. The similarity is calculated by inner product, which has a lower bound thus is conducive to optimization:

$$\mathcal{L}_{div} = \sum_{i=1}^{K} \left[ \ln \left( \sum_{j=1}^{K} (e^{<M_i, M_j>/\tau} \cdot \mathbb{1}_{i \neq j}) \right) \cdot scale \right] \qquad (5)$$

where $\tau$ is the temperature, $<x, y>$ denotes the inner product operation, $\mathbb{1}_{i \neq j}$ is the indicator function excludes the inner product of masking matrices with themselves, while $scale = 1/|M \ln 1/M|$ adjusts the numerical range of the diversity loss.

## 4 EXPERIMENTS

**Datasets** In our study, we select 20 commonly used tabular datasets in AD, which span diverse domains, including healthcare, finance, social sciences, etc. The datasets comprise 12 sourced from ODDS (Rayana, 2016) and 8 from ADBench (Han et al., 2022). Detailed statistical information for each dataset can be found in Table.18 of Appendix E.

**Evaluation metrics** We randomly partition the normal data of each dataset into two equal halves. The training dataset consists of one-half of the normal data, while the testing dataset comprises the other half of the normal data combined with all the abnormal instances. We employ Area Under the Receiver Operating Characteristic Curve (AUC-ROC) and Area Under the Precision-Recall Curve (AUC-PR) as our evaluation criteria.

**Implementation details** In our network architecture, the mask generator consists of a set of Multilayer Perceptron (MLP) attached by a sigmoid activation. The encoder and the decoder are symmetrical, both are three-layer MLP with LeakyReLU activation function. Our method is insensitive to hyperparameters, and the majority of them are not tuned for different datasets. In detail, we set epochs to 200, the batch size to 512, and fix the the number of masking matrices at 15. The temperature $\tau$ in diversity loss is set to 0.1. The number of hidden states of most layers in AE is 256, except for the bottleneck layer with the number of 128. A fixed set of hyperparameters is effective across all datasets, with dataset dimensions ranging widely from 6 to 500. The only two parameters tuned for different datasets are the learning rate and the weight $\lambda$ which balances two parts of loss. Adam optimizer is employed, bounded by an exponentially decaying learning rate controller.

**Baseline methods** We conduct a comparative analysis between MCM and nine other prominent methods in the field of tabular AD. Specifically, IForest (Liu et al., 2008), LOF (Breunig et al., 2000), OCSVM (Schölkopf et al., 1999), and ECOD (Li et al., 2022) represent classic AD approaches that continue to maintain popularity. The remaining five methods are deep learning based methods. Deep SVDD (Ruff et al., 2018) leverages NN to learn a hypersphere, and uses the distance between a sample and the center of hypersphere as anomaly score. DAGMM (Zong et al., 2018) combines an AE and Gaussian mixture model, and proposes a sample energy to judge anomalies. GOAD (Bergman & Hoshen, 2020), NeuTralAD (Qiu et al., 2021), and ICL (Shenkar & Wolf, 2022) are SSL based methods, which have been introduced in previous section. It is noteworthy that the implementation of five methods, including IForest, LOF, OCSVM, ECOD, and DeepSVDD, are sourced from the pyod (Zhao et al., 2019) library—a comprehensive Python library that integrates various AD methods. The implementation of the other four methods is based on their official open-source code releases. All the methods are implemented with identical dataset partitioning and preprocessing

Table 1: Comparison of AUC-PR results between baseline methods and MCM.

| Datset | IForset | LOF | OCSVM | ECOD | DeepSVDD | DAGMM | GOAD | NeuTralAD | ICL | Ours |
|---|---|---|---|---|---|---|---|---|---|---|
| Arrhythmia | 0.5097 | 0.5277 | 0.5339 | 0.4461 | 0.6036 | 0.4668 | 0.4091 | **0.6237** | 0.6155 | 0.6107 |
| Breastw | 0.9449 | 0.9923 | 0.9934 | 0.9522 | 0.9924 | 0.7584 | 0.8335 | 0.9117 | 0.9656 | **0.9952** |
| Cardio | 0.7018 | 0.8360 | **0.8614** | 0.3636 | 0.7880 | 0.3089 | 0.6225 | 0.4535 | 0.8037 | 0.8489 |
| Census | 0.1357 | 0.2343 | 0.2279 | 0.1773 | 0.1514 | 0.1066 | 0.1445 | 0.1103 | 0.1949 | **0.2420** |
| Campaign | 0.4608 | 0.4459 | 0.4749 | 0.4705 | 0.2529 | 0.2472 | 0.2030 | 0.4033 | 0.4506 | **0.6040** |
| Cardiotocography | 0.6036 | 0.5732 | 0.6619 | 0.6968 | 0.4602 | 0.444 | 0.4647 | 0.6431 | 0.5955 | **0.6993** |
| Fraud | **0.6939** | 0.4046 | 0.3490 | 0.4062 | 0.3320 | 0.0099 | 0.1249 | 0.2312 | 0.5960 | 0.5141 |
| Glass | 0.0952 | 0.0923 | 0.0896 | 0.1113 | 0.0912 | 0.1019 | 0.0948 | 0.1491 | **0.2573** | 0.1905 |
| Ionosphere | 0.9768 | 0.9591 | 0.8969 | 0.9713 | 0.8670 | 0.7046 | 0.9280 | 0.9355 | 0.9777 | **0.9802** |
| Mammography | 0.3334 | 0.4063 | 0.4178 | **0.5380** | 0.4190 | 0.1141 | 0.2426 | 0.1326 | 0.1894 | 0.4755 |
| NSL-KDD | 0.7532 | 0.7450 | 0.7529 | 0.6355 | 0.7935 | 0.7504 | 0.7823 | 0.8385 | 0.5194 | **0.9085** |
| Optdigits | 0.157 | 0.4363 | 0.0692 | 0.0669 | 0.1159 | 0.0536 | 0.0633 | 0.1709 | 0.1696 | **0.8885** |
| Pima | 0.6662 | 0.6970 | 0.7008 | 0.5877 | 0.7165 | 0.5956 | 0.5027 | 0.6168 | 0.6965 | **0.7389** |
| Pendigits | 0.5133 | 0.7855 | 0.5178 | 0.4145 | 0.0616 | 0.0441 | 0.0259 | 0.693 | 0.4039 | **0.8258** |
| Satellite | 0.8583 | 0.8088 | 0.7778 | 0.8334 | 0.8217 | 0.6866 | 0.7786 | 0.8588 | **0.8799** | 0.8532 |
| Satimage-2 | 0.8846 | 0.9692 | 0.9192 | 0.7775 | 0.9427 | 0.1142 | 0.9726 | 0.9684 | 0.8124 | **0.9850** |
| Shuttle | 0.9172 | 0.9601 | 0.9488 | 0.9815 | 0.9818 | 0.4875 | 0.9545 | **0.9971** | 0.9811 | 0.9479 |
| Thyroid | 0.6055 | 0.7892 | 0.8134 | 0.6807 | 0.7282 | 0.1095 | 0.6931 | 0.7435 | 0.6575 | **0.8417** |
| Wbc | 0.8573 | 0.8412 | 0.8391 | 0.7217 | 0.8340 | 0.2959 | 0.3362 | 0.6051 | 0.7218 | **0.8887** |
| Wine | 0.2458 | 0.1253 | 0.1424 | 0.3578 | 0.1476 | 0.4907 | 0.1619 | 0.9078 | 0.5659 | **0.9335** |
| Average | 0.5957 | 0.6315 | 0.5994 | 0.5595 | 0.5551 | 0.3445 | 0.4669 | 0.5997 | 0.6027 | **0.7486** |

procedures, following the latest works (Qiu et al., 2021; Shenkar & Wolf, 2022). Every experiment runs three times and we report the mean results throughout this paper.

## 4.1 MAIN RESULTS

Table 1 depicts the performance of MCM in relation to other methods. Due to space constraints, we exclusively show the AUC-PR results in the table. AUC-ROC performance can be found in Table.5 in Appendix A, which is generally consistent with the AUC-PR performance. As is shown in Table 1, MCM achieves the best performance on 13 out of the 20 datasets. Notably, MCM outperforms other methods significantly on various datasets, such as NSL-KDD. Even on datasets where it falls slightly short of the best-performing method, MCM's performance remains commendable and the performance gap remains within an acceptable range. As for average results, MCM obtains an absolute gain of at least 11.7% compared to other methods. These evaluation results demonstrate the effectiveness of MCM.

## 4.2 ABLATION STUDIES

In this subsection, we conduct ablation study to evaluate the effectiveness of individual components of our network. We calculate the AUC-ROC and AUC-PR on all the datasets above, and show the average results in Table 2. The detailed results for each dataset are provided in Table.6 and Table.7 of Appendix B.

- **Task A**: We use a vanilla AE without applying any masking as a baseline to be compared.
- **Task B**: Compared with our approach (Task E), Task B replaces learnable masking matrices with random sampled masking matrices, which leads to significant decrease in performance. Without a suitable masking strategy, the results are even worse than AE which is without a masking operation, proving the important role of learnable masking strategy.
- **Task C**: We do not use the ensemble learning method and set the number of masks to one, so diversity loss is also inapplicable. The performance gap between Task C and Task E indicates that an ensemble of multiple and diverse masks can enhance detection capabilities.
- **Task D**: Compared to Task E, Task D lacks of diversity loss and leads to inferior results, highlighting the importance of diversity loss. It is worth noting that the results of Task D and Task C are similar. Compared to Task C, Task D uses ensemble learning but without diversity loss. This suggests that simply increasing the number of masks doesn't work, as the model tends to learn redundant masks. The interaction between ensemble and diversity is necessary to capture diverse correlations which can characterize normal data better.

- **Task E**: Our method outperforms other variants by a considerable gap, validating the effectiveness of each component of MCM.

## 4.3 DIFFERENT MASKING STRATEGIES

As the first work to extend the success of MIM/MLM to tabular AD, we systematically study different options for masking strategy.

- **Matrix Masking**: Generate a matrix $M$ whose values are randomly drawn from a uniform distribution, and then mask all the input by performing element-wise product with $M$.
- **Zero Masking**: Randomly sample some features with probability $p_m$, and mask them with zero value.
- **Transformation Masking**: Generate a matrix $W$ whose values are randomly drawn from a normal distribution, and then perform matrix multiplication with input and $W$, as proposed in (Bergman & Hoshen, 2020).
- **Shuffle Masking**: Randomly sample some features with probability $p_m$, and mask them with a random draw from that feature's empirical marginal distribution, as proposed in (Bahri et al., 2022).
- **Gumbel Masking**: Adopt Gumbel-Softmax (Jang et al., 2017) instead of sigmoid function in mask generator G for generating learnable binary masks.

In our experiments, the masking ratio $p_m$ of zero and shuffle masking is set to 0.4. In Table 3, we present the average performance of different masking strategies (see Appendix B for the table of 20 datasets). The results show that our learnable masking strategy remains the top performer on the majority of datasets. Our average AUC-PR outperforms the second-ranked strategy with an absolute value of 8.44%, which is quite impressive. Above results further demonstrate that masks produced randomly in other strategies are not effective enough. For zero and shuffle masking, the randomness is reflected in the selection of features to be masked, while for

Table 2: Ablation study of MCM and its variants.

| Task | Learnable | Ensemble | $\mathcal{L}_{div}$ | AUCROC | AUCPR |
|------|-----------|----------|---------------------|--------|-------|
| A | - | - | - | 0.8732 | 0.6849 |
| B | - | ✓ | - | 0.8367 | 0.6178 |
| C | ✓ | - | - | 0.8883 | 0.7034 |
| D | ✓ | ✓ | - | 0.8898 | 0.7029 |
| E | ✓ | ✓ | ✓ | **0.9044** | **0.7486** |

Table 3: Average AUC-ROC and AUC-PR results of different masking strategies.

| Metrics | Matrix | Zero | Trans. | Shuffle | Gumbel | Ours |
|---------|--------|------|--------|---------|--------|------|
| AUCROC | 0.8367 | 0.8497 | 0.8720 | 0.8490 | 0.8036 | **0.9044** |
| AUCPR | 0.6178 | 0.6611 | 0.6642 | 0.6149 | 0.6012 | **0.7486** |

matrix and transformation masking, the randomness is introduced by the generation of $M$ and $W$. Such random strategies make it easy to produce meaningless masks for tabular AD. Different from them, our masks are generated based on data and introduce no randomness, which can capture characteristic correlations in normal data and contribute to detecting anomalies.

## 4.4 DIFFERENT TYPES OF ANOMALIES

Although the types of anomalies are infinite, existing works (Steinbuss & Böhm, 2021; Han et al., 2022) have summarized four common types of anomalies and proposed methods for generating them. Following the same setup, we conduct experiments to generate synthetic anomalies based on a realistic Shuttle dataset and assess the performance of MCM on specific types of anomalies. The generation process of four types of anomalies is described in Appendix C. A brief introduction of four types of anomalies is as follows:

- **Local anomalies** refer to the anomalies that are deviant from their local neighborhoods.
- **Global anomalies** are samples scattering throughout the entire space, while being far from the normal data distribution.
- **Dependency anomalies** represent samples that do not adhere to the correlation structure among normal data.
- **Clustered anomalies** refer to groups of data with similar characteristics but significantly different from normal data.

Table 4 displays the performance of MCM and competitive methods on specific types of anomalies. MCM exhibits strong performance across all anomaly types. For cluster anomalies and global anomalies, nearly all the methods achieve excellent results as these two types are relatively simple

Table 4: Results of our method and baselines on different types of anomalies. We initially build a distribution by fitting it to the training data of the Shuttle dataset. Normal samples generated are directly sampled from this distribution, while abnormal samples generated are obtained from distributions that are made specific alterations for certain anomaly types.

| | Metrics | IForest | LOF | OCSVM | ECOD | DeepSVDD | DAGMM | GOAD | NeuTralAD | ICL | Ours |
|---|---|---|---|---|---|---|---|---|---|---|---|
| **Local** | AUC-ROC | 0.8322 | 0.8701 | 0.8137 | 0.8391 | 0.6400 | 0.6895 | 0.2975 | 0.8276 | 0.6851 | **0.9226** |
| | AUC-PR | 0.3494 | **0.5256** | 0.3847 | 0.4324 | 0.2932 | 0.1461 | 0.0822 | 0.4247 | 0.3284 | 0.5103 |
| **Cluster** | AUC-ROC | 0.9995 | 0.9996 | 0.9996 | 0.9955 | 0.9996 | 0.9138 | 0.9999 | 1.0000 | 1.0000 | **1.0000** |
| | AUC-PR | 0.9888 | 0.9772 | 0.9909 | 0.9272 | 0.9772 | 0.3087 | 0.9885 | 0.9995 | 1.0000 | **1.0000** |
| **Dependency** | AUC-ROC | 0.7821 | 0.8929 | 0.8464 | 0.8929 | 0.8750 | 0.8750 | 0.3929 | 0.6071 | 0.8036 | **0.9107** |
| | AUC-PR | 0.6333 | 0.3250 | 0.5000 | 0.6250 | 0.3095 | 0.2222 | 0.0913 | 0.1213 | 0.3333 | **0.6429** |
| **Global** | AUC-ROC | 0.9999 | 0.9976 | 0.9994 | 1.0000 | 0.9999 | 0.9074 | 0.9994 | 0.9999 | 0.9998 | **1.0000** |
| | AUC-PR | 0.9992 | 0.9453 | 0.9847 | 0.9996 | 0.9990 | 0.2936 | 0.9956 | 0.9990 | 0.9992 | **1.0000** |

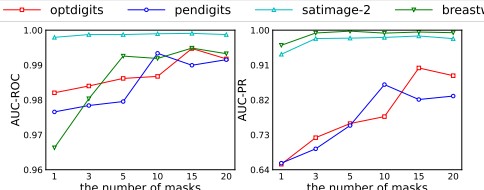

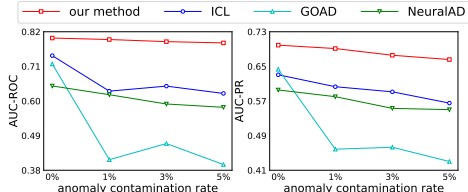

Figure 3: AUC-ROC and AUC-PR performance of parameter sensitivity analysis on the number of masking matrices.

Figure 4: AUC-ROC and AUC-PR performance on Cardiotocography dataset with different ratio of anomaly contamination.

to detect. In the case of local anomalies, our method performs just behind LOF, which is specifically designed and tailored to detect local anomalies. For dependency anomalies, our method shows a leading performance, because MCM can extract correlations in normal data and is thus skilled in detecting anomalies that lack correlations between features.

### 4.5 FURTHER ANALYSIS

**Robustness to Anomaly Contamination** Our method is implemented in a one-class classification setting where all the samples in training set are normal data. In some real-world applications, the training set may also contain a small percentage of anomalies, i.e., anomaly contamination. To analyse the robustness of MCM w.r.t anomaly contamination, we conduct experiments in the case of anomaly contamination ratio of 0%, 1%, 3%, and 5%. Here we only show the results of Cardiotocography dataset, see Appendix D for results of several other datasets. From Figure.4, we can observe that all methods suffer from a performance decline as the contamination rate increases. But compared to the other three SSL based methods, MCM performs more stable and consistently shows the best performance, demonstrating a superior robustness of MCM to anomaly contamination.

**The Number of Masking Matrices** Figure.3 shows the performance of MCM with a varying number of masking matrices. At the beginning, the performance on all the four datasets has an obvious improvement as the number of masking matrices increases. Because more diverse masking matrices can extract more characteristic correlations of normal data and provide more discriminative evidence to judge anomalies. When the number of masking matrices comes to a relatively high value, the performance becomes stable. In MCM, we fix this parameter to 15 and don't adjust it for different datasets. The sensitivity experiments of other hyperparameters are provided in Appendix B.

## 5 DISCUSSION

In this section, due to the space limitation, we mainly discuss the interpretability of MCM here, see Appendix D for other discussions including lightweight decoder and pure masked prediction, Appendix G for visualization of masks, and Appendix H for discussion about mask degeneration. MCM can provide interpretability from two aspects: correlations between features and abnormality of each individual feature. Note the calculation of anomaly score of each sample has two averaging

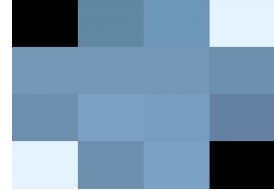

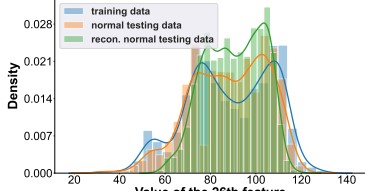

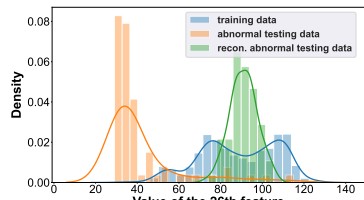

Figure 5: Visualization of masks on one anomaly. The darker/lighter color indicates a value near 0/1.

Figure 6: KDE of 26th feature of Satimage-2. The left/right figure shows the distribution of normal/abnormal testing data before and after reconstruction, both are compared with the distribution of training data.

operations. The first is averaged on different features and the second is averaged on different masked versions. We define average reconstruction error from one masked version across different features as **per-mask contribution**, and define average reconstruction error of one feature across different masked versions as **per-feature contribution**.

**Correlations between features** are represented by different masks in MCM. Per-mask contribution can give interpretability by pointing out a sample deviates from which characteristic correlations exist in normal data. Referred to (Shenkar & Wolf, 2022), we conduct a case study. Normal data are vectors over $\mathbb{R}^4$ by sampling from a Gaussian distribution. The covariance matrix shows a high covariance of 0.85 between the features of indices 1 and 4, while the covariance is zero elsewhere. Anomalies are sampled from similar distributions with no correlation between any of the features.

For the convenience of display, we set the number of masks to four. We randomly sample one anomaly and visualize its masks in Fig.5. Meanwhile, we calculate its corresponding per-mask contribution: [**39%**, 14%, 7%, **40%**]. In this case, the 1st and the 4th masks contribute the most. Refer to Fig.5, these two masks mainly reserve the 4th/1st feature and mask the 1st/4th feature. So a higher contribution of these two masks illustrates the correlation between the 1st and the 4th dimensional features of this sample is distinct from that of normal data.

**Each individual feature** In Fig.6, We draw KDE of 26th feature of Satimage-2 dataset. The left/right figure shows the distribution of normal/abnormal testing data before and after reconstruction, reconstructed data used are the average of all masked versions. As is shown, the distribution of normal testing data before and after reconstruction are of little difference, both are basically identical with that of training data. However, for abnormal testing data, the distribution after reconstruction shifts significantly towards training data. Because MCM accesses to only normal data when training, it will implicitly learn the empirical marginal distribution of each feature of normal data and prefers to reconstruct masked features to this distribution, thus abnormal samples whose distribution is far from training data will have larger reconstruction errors compared to normal samples.

Based on the above qualitative analysis, we conduct a quantitative study on Thyroid dataset, which has six features: [TSH, T3, TT4, T4U, FT1, TBG]. We calculate average per-feature contribution on all the anomalies and obtain the corresponding results: [**16%**, **43%**, 1%, 9%, 11% , 9%]. The two highest per-feature contributions are from "T3" (43%) and "TSH" (16%). These results hold practical significance in medical disease detection since hyperthyroidism is characterized by elevated "T3" levels and decreased "TSH" levels. Therefore, per-feature contribution can complement per-mask contribution and provide interpretability from the view of individual features.

# 6 CONCLUSION

Inspired by the success of MIM/MLM in the CV and NLP domains, we present MCM framework for anomaly detection in tabular data. We emphasize the significance of masking strategy for tabular AD and propose to generate masks by learning. Moreover, we design a loss to enhance the diversity of different masks, which contributes to extracting diverse characteristic correlations in normal data, thus anomalies can be distinguished by a lack of such correlations. Our method achieves state-of-the-art performance on extensive experiments and provides interpretability from the view of individual feature and the correlations between different features.

## 7    ACKNOWLEDGMENTS

This work is supported in part by the National Natural Science Foundation of China (No. 62272057), and the Major Science and Technology Projects in Anhui Province (No.202203a05020025).

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

## A    ADDITIONAL RESULTS

In Table.5 we display the AUC-ROC performance of MCM and comparing methods.

## B    ABLATIONS AND PARAMETER SENSITIVITY

In Table.6 and Table.7, we present the detailed AUC-ROC and AUC-PR performance of ablation study on all selected datasets.

Table.8 and Table.9 show the AUC-ROC and AUC-PR results on different masking strategies.

Table.10 and Table.11 show the AUC-ROC and AUC-PR results with different weight coefficient $\lambda$. In the previous experiments, we fine-tuned $\lambda$ for each dataset. Here, to explore the sensitivity to $\lambda$, we conducted experiments using a fixed $\lambda$ value across all datasets. The results indicate that $\lambda = 20$ achieves best performance. More importantly, in comparison to that $\lambda$ is adjusted individually for each dataset, the average AUCPR only decreased by 0.0231, which still surpasses all baseline methods, proving that our method is not sensitive to $\lambda$.

Table 5: Comparison of AUC-ROC results between baseline methods and MCM.

| Dataset | IForest | LOF | OCSVM | ECOD | DeepSVDD | DAGMM | GOAD | NeuTralAD | ICL | Ours |
|---------|---------|-----|-------|------|----------|-------|------|-----------|-----|------|
| Arrhythmia | 0.7734 | 0.7688 | 0.7689 | 0.7199 | 0.7941 | 0.7283 | 0.5810 | 0.7900 | **0.8145** | 0.8114 |
| Breastw | 0.9719 | 0.9937 | 0.9938 | 0.9649 | 0.9925 | 0.7605 | 0.7327 | 0.9458 | 0.9725 | **0.9955** |
| Cardio | 0.9220 | 0.9562 | **0.9654** | 0.6370 | 0.9313 | 0.6951 | 0.6812 | 0.7349 | 0.9514 | 0.9603 |
| Census | 0.6015 | 0.7110 | 0.7158 | 0.5772 | 0.6327 | 0.4506 | 0.5615 | 0.4685 | 0.6831 | 0.7581 |
| Campaign | 0.7269 | 0.7061 | 0.7626 | 0.7555 | 0.5254 | 0.5785 | 0.453 | 0.6982 | 0.7241 | **0.8902** |
| Cardiotocography | 0.7249 | 0.6449 | 0.7522 | 0.7889 | 0.5149 | 0.6101 | 0.4428 | 0.7177 | 0.6478 | **0.8001** |
| Fraud | **0.9630** | 0.9573 | 0.9548 | 0.8535 | 0.9519 | 0.7271 | 0.4808 | 0.8807 | 0.9107 | 0.9357 |
| Glass | 0.5771 | 0.5620 | 0.5480 | 0.6235 | 0.5566 | 0.5982 | 0.5741 | 0.6312 | **0.8350** | 0.7225 |
| Ionosphere | 0.9683 | 0.9454 | 0.8765 | 0.9569 | 0.8552 | 0.7041 | 0.8993 | 0.9453 | 0.9710 | **0.9726** |
| Mammography | 0.8220 | 0.8922 | 0.9003 | 0.8251 | 0.8879 | 0.7412 | 0.7348 | 0.7449 | 0.6548 | **0.9053** |
| NSL-KDD | 0.7388 | 0.5496 | 0.5707 | 0.3810 | 0.7551 | 0.6136 | 0.6575 | 0.7701 | 0.1684 | **0.8780** |
| Optdigits | 0.8239 | 0.9665 | 0.6338 | 0.6145 | 0.7603 | 0.4706 | 0.5972 | 0.8471 | 0.787 | **0.9947** |
| Pima | 0.6737 | 0.6913 | 0.7133 | 0.5834 | 0.7348 | 0.6108 | 0.4338 | 0.6170 | 0.6727 | **0.7639** |
| Pendigits | 0.9666 | 0.9905 | 0.9636 | 0.9295 | 0.4563 | 0.3982 | 0.2141 | 0.9859 | 0.9142 | **0.9919** |
| Satellite | 0.8026 | 0.7391 | 0.6663 | 0.7884 | 0.7659 | 0.7259 | 0.7374 | 0.8080 | **0.8549** | 0.7962 |
| Satimage-2 | 0.9938 | 0.9961 | 0.9817 | 0.9650 | 0.9881 | 0.8994 | 0.9929 | 0.9979 | 0.9792 | **0.9992** |
| Shuttle | 0.9961 | 0.9983 | 0.9969 | 0.9978 | 0.9952 | 0.9049 | 0.9897 | **0.9994** | 0.9935 | 0.9975 |
| Thyroid | 0.9271 | 0.9856 | 0.9855 | 0.8827 | **0.9887** | 0.7149 | 0.8523 | 0.9701 | 0.9518 | 0.9804 |
| Wbc | 0.9715 | 0.9670 | 0.9667 | 0.8747 | 0.9633 | 0.7999 | 0.4806 | 0.9133 | 0.9080 | **0.9814** |
| Wine | 0.6571 | 0.4083 | 0.485 | 0.7433 | 0.5067 | 0.8833 | 0.5366 | **0.9753** | 0.915 | 0.9538 |
| Avergae | 0.8301 | 0.8215 | 0.8101 | 0.7731 | 0.7778 | 0.6808 | 0.6317 | 0.8221 | 0.8155 | **0.9044** |

Table.12 and Table.13 show the AUC-ROC and AUC-PR results with different batch size.

Table.14 and Table.15 show the AUC-ROC and AUC-PR results with different diversity loss designs. When running and testing on the same dataset, we ensure that the weight coefficient $\lambda$ remains constant.

Figure.7 shows the visualization of parameter sensitivity analysis on the weight coefficient $\lambda$, the number of epochs, and the batch size. The result is consistent with section.4.5 that our model is generally insensitive to parameters.

In Figure.8, we visualize the AUC-ROC and AUC-PR performance with different ratios of anomaly contamination on four datasets: Breastw, Cardio, Cardiotocography, and Wbc. As is mentioned in section4.5, our method presents strong robustness compared to the other three self-supervised methods.

Table 6: AUC-ROC results for ablation study on five sets of tasks.

| Dataset | TASK A | TASK B | TASK C | TASK D | TASK E |
|---------|--------|--------|--------|--------|--------|
| Arrhythmia | 0.7793 | 0.7697 | 0.7848 | 0.8036 | **0.8114** |
| Breastw | 0.9579 | 0.9904 | 0.9702 | 0.9488 | **0.9955** |
| Campaign | 0.8916 | 0.7312 | **0.8927** | 0.8907 | 0.8902 |
| Cardio | 0.9429 | **0.9697** | 0.9370 | 0.9361 | 0.9603 |
| Cardiotocography | 0.7686 | 0.7893 | 0.7860 | 0.7906 | **0.8001** |
| Census | 0.7443 | 0.6834 | 0.7360 | 0.7455 | **0.7581** |
| Fraud | 0.8977 | 0.9110 | **0.9361** | 0.9252 | 0.9357 |
| Glass | 0.7167 | 0.6029 | **0.7225** | 0.7000 | **0.7225** |
| Ionosphere | 0.9680 | 0.7689 | 0.9680 | **0.9731** | 0.9726 |
| Mammography | 0.7969 | 0.8492 | 0.8661 | 0.8801 | **0.9053** |
| NSL-KDD | 0.8748 | 0.8269 | 0.8695 | 0.8747 | **0.8780** |
| Optdigits | 0.9629 | 0.9509 | 0.9826 | 0.9761 | **0.9947** |
| Pendigits | 0.9885 | 0.9588 | 0.9841 | 0.9899 | **0.9919** |
| Pima | 0.6023 | 0.6802 | 0.6722 | 0.7131 | **0.7639** |
| Satellite | 0.7969 | 0.6267 | 0.7975 | 0.7960 | **0.7962** |
| Satimage-2 | 0.9980 | 0.9960 | 0.9985 | 0.9987 | **0.9992** |
| Shuttle | 0.9990 | 0.9955 | 0.9986 | **0.9983** | 0.9975 |
| Thyroid | 0.9795 | 0.8125 | 0.9804 | **0.9826** | 0.9804 |
| Wbc | 0.8376 | 0.9017 | 0.9408 | 0.9408 | **0.9814** |
| Wine | **0.9599** | 0.9199 | 0.9414 | 0.9322 | 0.9538 |
| Average | 0.8732 | 0.8367 | 0.8883 | 0.8898 | **0.9044** |

Table 7: AUC-PR results for Ablation study on five sets of tasks.

| Dataset | TASK A | TASK B | TASK C | TASK D | TASK E |
|---|---|---|---|---|---|
| Arrhythmia | 0.5700 | 0.5596 | 0.5869 | 0.6024 | **0.6107** |
| Breastw | 0.9567 | 0.9868 | 0.9608 | 0.9373 | **0.9952** |
| Campaign | **0.6128** | 0.5022 | 0.6088 | 0.6003 | 0.6040 |
| Cardio | 0.7756 | **0.8569** | 0.6315 | 0.6209 | 0.8489 |
| Cardiotocography | 0.6685 | 0.6689 | 0.7006 | 0.6918 | **0.6993** |
| Census | 0.2184 | 0.1798 | 0.2151 | 0.2252 | **0.2420** |
| Fraud | 0.5294 | 0.3978 | **0.5346** | 0.5032 | 0.5141 |
| Glass | 0.1904 | 0.1605 | **0.3178** | 0.1566 | 0.1905 |
| Ionosphere | 0.9778 | 0.7819 | 0.9773 | **0.9806** | 0.9802 |
| Mammography | 0.2589 | 0.2970 | 0.3150 | 0.3310 | **0.4755** |
| NSL-KDD | 0.9051 | 0.8643 | 0.8975 | 0.9030 | **0.9085** |
| Optdigits | 0.5582 | 0.4794 | 0.6300 | 0.6281 | **0.8885** |
| Pendigits | 0.7805 | 0.4392 | 0.6959 | 0.7826 | **0.8258** |
| Pima | 0.6240 | 0.6962 | 0.6551 | 0.7013 | **0.7389** |
| Satellite | 0.8542 | 0.7183 | 0.8544 | 0.8532 | **0.8532** |
| Satimage-2 | 0.9273 | 0.9663 | 0.9505 | 0.9798 | **0.9850** |
| Shuttle | 0.9856 | 0.9293 | 0.9683 | **0.9667** | 0.9479 |
| Thyroid | 0.8362 | 0.3332 | **0.8457** | 0.8355 | 0.8417 |
| Wbc | 0.5841 | 0.6132 | 0.8464 | 0.8464 | **0.8887** |
| Wine | 0.8834 | 0.9250 | 0.8758 | 0.9128 | **0.9335** |
| Average | 0.6849 | 0.6178 | 0.7034 | 0.7029 | **0.7486** |

Table 8: AUC-ROC results on different masking strategies.

| Datset | Matirx | Zero | Transformation | Shuffle | Gumbel | Ours |
|---|---|---|---|---|---|---|
| Arrhythmia | 0.7697 | 0.7895 | 0.7707 | **0.8146** | 0.7257 | 0.8114 |
| Breastw | 0.9904 | 0.9875 | 0.9891 | 0.9836 | 0.9918 | **0.9955** |
| Campaign | 0.7312 | 0.6751 | 0.8007 | 0.6999 | 0.4557 | **0.8902** |
| Cardio | **0.9697** | 0.9459 | 0.9298 | 0.9394 | 0.9531 | 0.9603 |
| Cardiotocography | 0.7893 | 0.7860 | 0.7924 | **0.8206** | 0.7012 | 0.8001 |
| Census | 0.6834 | 0.6470 | 0.6884 | 0.7173 | **0.8199** | 0.7581 |
| Fraud | 0.9110 | 0.9427 | 0.9417 | **0.9851** | 0.8998 | 0.9357 |
| Glass | 0.6029 | 0.4824 | 0.6441 | 0.5843 | 0.7151 | **0.7225** |
| Ionosphere | 0.7689 | 0.9639 | 0.9703 | 0.9709 | 0.7845 | **0.9726** |
| Mammography | 0.8492 | 0.8571 | 0.8361 | 0.8410 | 0.8802 | **0.9053** |
| NSL-KDD | 0.8269 | 0.7684 | 0.8638 | 0.6458 | 0.6575 | **0.8780** |
| Optdigits | 0.9509 | 0.9842 | 0.9840 | 0.9071 | 0.2493 | **0.9947** |
| Pendigits | 0.9588 | 0.9809 | 0.9769 | 0.9709 | 0.9318 | **0.9919** |
| Pima | 0.6802 | 0.6760 | 0.6819 | 0.5416 | 0.7187 | **0.7639** |
| Satellite | 0.6267 | 0.7762 | 0.7901 | **0.8559** | 0.6415 | 0.7962 |
| Satimage-2 | 0.9960 | 0.9979 | **0.9992** | 0.9986 | 0.9908 | **0.9992** |
| Shuttle | 0.9955 | 0.9966 | 0.9971 | 0.9909 | 0.9926 | **0.9975** |
| Thyroid | 0.8125 | 0.8952 | 0.9580 | 0.8622 | 0.9679 | **0.9804** |
| Wbc | 0.9017 | 0.9285 | 0.9030 | 0.9413 | 0.8989 | **0.9814** |
| Wine | 0.9199 | 0.9137 | 0.9230 | 0.9091 | 0.9461 | **0.9538** |
| Average | 0.8367 | 0.8497 | 0.8720 | 0.8490 | 0.8036 | **0.9044** |

Table 9: AUC-PR results on different masking strategies.

| Dataset | Matrix | Zero | Transformation | Shuffle | Gumbel | Ours |
|---|---|---|---|---|---|---|
| Arrhythmia | 0.5596 | 0.5891 | 0.5724 | **0.6113** | 0.5530 | 0.6107 |
| Breastw | 0.9868 | 0.9820 | 0.9865 | 0.9837 | 0.9919 | **0.9952** |
| Campaign | 0.5022 | 0.3760 | 0.5299 | 0.3623 | 0.1852 | **0.6040** |
| Cardio | **0.8569** | 0.8329 | 0.7718 | 0.7473 | 0.8380 | 0.8489 |
| Cardiotocography | 0.6689 | 0.7006 | 0.6901 | **0.7076** | 0.6106 | 0.6993 |
| Census | 0.1798 | 0.1633 | 0.1795 | 0.2103 | **0.3853** | 0.2420 |
| Fraud | 0.3978 | 0.4845 | 0.4886 | **0.5384** | 0.1810 | 0.5141 |
| Glass | 0.1605 | 0.1109 | 0.1326 | 0.1588 | 0.1900 | **0.1905** |
| Ionosphere | 0.7819 | 0.9758 | 0.9789 | 0.9806 | 0.8523 | **0.9802** |
| Mammography | 0.2970 | 0.3210 | 0.3085 | 0.2223 | 0.3082 | **0.4755** |
| NSL-KDD | 0.8643 | 0.8577 | 0.8971 | 0.8034 | 0.7823 | **0.9085** |
| Optdigits | 0.4794 | 0.7172 | 0.7434 | 0.2788 | 0.0342 | **0.8885** |
| Pendigits | 0.4392 | 0.6142 | 0.5039 | 0.4850 | 0.3628 | **0.8258** |
| Pima | 0.6962 | 0.6743 | 0.6730 | 0.5817 | 0.7131 | **0.7389** |
| Satellite | 0.7183 | 0.8308 | 0.8411 | **0.8661** | 0.7198 | 0.8532 |
| Satimage-2 | 0.9663 | 0.9718 | 0.9833 | 0.9800 | 0.9633 | **0.9850** |
| Shuttle | 0.9293 | 0.9473 | 0.9442 | 0.8536 | 0.9201 | **0.9479** |
| Thyroid | 0.3332 | 0.5106 | 0.6772 | 0.3618 | 0.8167 | **0.8417** |
| Wbc | 0.6132 | 0.6764 | 0.4643 | 0.7753 | 0.6183 | **0.8887** |
| Wine | 0.9250 | 0.8861 | 0.9175 | 0.7895 | 0.9230 | **0.9335** |
| Average | 0.6178 | 0.6611 | 0.6642 | 0.6149 | 0.6012 | **0.7486** |

Table 10: AUC-ROC results with different weight coefficient $\lambda$.

| DataSet | $\lambda=1$ | $\lambda=5$ | $\lambda=10$ | $\lambda=20$ | $\lambda=50$ | $\lambda=100$ |
|---|---|---|---|---|---|---|
| Arrhythmia | 0.8078 | 0.8114 | 0.8089 | 0.8103 | 0.8112 | 0.8101 |
| Breastw | 0.9832 | 0.9952 | 0.9955 | 0.9956 | 0.9956 | 0.9957 |
| Campaign | 0.8603 | 0.8603 | 0.8634 | 0.8617 | 0.8660 | 0.8759 |
| Cardio | 0.7767 | 0.8132 | 0.8751 | 0.9205 | 0.9468 | 0.9603 |
| Cardiotocography | 0.8001 | 0.8024 | 0.7947 | 0.7929 | 0.7966 | 0.8019 |
| Census | 0.7581 | 0.7516 | 0.6865 | 0.7375 | 0.7095 | 0.7322 |
| Fraud | 0.9357 | 0.9343 | 0.9322 | 0.9303 | 0.9268 | 0.9236 |
| Glass | 0.6382 | 0.7284 | 0.7118 | 0.6931 | 0.6951 | 0.7225 |
| Ionosphere | 0.9594 | 0.9622 | 0.9726 | 0.9629 | 0.9633 | 0.9636 |
| Mammography | 0.8636 | 0.8697 | 0.8758 | 0.8760 | 0.8947 | 0.9053 |
| Nslkdd | 0.8744 | 0.8675 | 0.8715 | 0.8780 | 0.8701 | 0.8788 |
| Optdigits | 0.9792 | 0.9937 | 0.9947 | 0.9945 | 0.9856 | 0.9856 |
| Pendigits | 0.9858 | 0.9900 | 0.9912 | 0.9930 | 0.9870 | 0.9860 |
| Pima | 0.7639 | 0.7289 | 0.7041 | 0.7243 | 0.7363 | 0.6866 |
| Satellite | 0.7950 | 0.7962 | 0.7945 | 0.7958 | 0.7944 | 0.7944 |
| Satimage-2 | 0.9990 | 0.9992 | 0.9992 | 0.9989 | 0.9987 | 0.9986 |
| Shuttle | 0.9975 | 0.9973 | 0.9975 | 0.9975 | 0.9974 | 0.9973 |
| Thyroid | 0.9804 | 0.9639 | 0.9514 | 0.9472 | 0.9371 | 0.9323 |
| Wbc | 0.9791 | 0.9814 | 0.9724 | 0.9594 | 0.9244 | 0.9298 |
| Wine | 0.9492 | 0.9445 | 0.9414 | 0.9507 | 0.9538 | 0.9245 |
| Average | 0.8843 | 0.8896 | 0.8867 | 0.8910 | 0.8895 | 0.8903 |

Table 11: AUC-PR results with different weight coefficient $\lambda$.

| DataSet | $\lambda$=1 | $\lambda$=5 | $\lambda$=10 | $\lambda$=20 | $\lambda$=50 | $\lambda$=100 |
|---|---|---|---|---|---|---|
| Arrhythmia | 0.6064 | 0.6107 | 0.6050 | 0.6079 | 0.6152 | 0.6084 |
| Breastw | 0.9753 | 0.9947 | 0.9952 | 0.9954 | 0.9954 | 0.9955 |
| Campaign | 0.5547 | 0.5540 | 0.5570 | 0.5551 | 0.5600 | 0.5760 |
| Cardio | 0.3372 | 0.3986 | 0.5615 | 0.7831 | 0.8146 | 0.8489 |
| Cardiotocography | 0.6993 | 0.7006 | 0.6936 | 0.6907 | 0.6941 | 0.6976 |
| Census | 0.2420 | 0.2280 | 0.1807 | 0.2099 | 0.1907 | 0.2234 |
| Fraud | 0.5141 | 0.5065 | 0.4911 | 0.4688 | 0.4404 | 0.4208 |
| Glass | 0.1218 | 0.1798 | 0.2265 | 0.1865 | 0.1556 | 0.1905 |
| Ionosphere | 0.9724 | 0.9740 | 0.9802 | 0.9744 | 0.9747 | 0.9748 |
| Mammography | 0.3903 | 0.3623 | 0.3672 | 0.4619 | 0.4858 | 0.4755 |
| Nslkdd | 0.9029 | 0.8985 | 0.8975 | 0.9085 | 0.8988 | 0.9032 |
| Optdigits | 0.6443 | 0.8723 | 0.8885 | 0.8851 | 0.7523 | 0.7118 |
| Pendigits | 0.7326 | 0.8164 | 0.8065 | 0.8168 | 0.6938 | 0.6613 |
| Pima | 0.7389 | 0.7019 | 0.6800 | 0.6967 | 0.7075 | 0.6922 |
| Satellite | 0.8520 | 0.8532 | 0.8503 | 0.8528 | 0.8514 | 0.8496 |
| Satimage-2 | 0.9836 | 0.9845 | 0.9850 | 0.9808 | 0.9788 | 0.9778 |
| Shuttle | 0.9479 | 0.9440 | 0.9456 | 0.9456 | 0.9446 | 0.9409 |
| Thyroid | 0.8417 | 0.8192 | 0.8004 | 0.7927 | 0.7521 | 0.7340 |
| Wbc | 0.8743 | 0.8887 | 0.8364 | 0.7731 | 0.7381 | 0.7233 |
| Wine | 0.9241 | 0.9226 | 0.9217 | 0.9246 | 0.9335 | 0.9178 |
| Average | 0.6928 | 0.7105 | 0.7135 | 0.7255 | 0.7089 | 0.7062 |

Table 12: AUC-ROC results with different batch size.

| DataSet | batch size=32 | batch size=64 | batch size=128 | batch size=256 | batch size=512 |
|---|---|---|---|---|---|
| Arrhythmia | 0.7819 | 0.7690 | 0.7789 | 0.8113 | 0.8114 |
| Breastw | 0.9901 | 0.9939 | 0.9966 | 0.9955 | 0.9955 |
| Campaign | 0.7801 | 0.7907 | 0.8133 | 0.8271 | 0.8902 |
| Cardio | 0.8689 | 0.8897 | 0.9011 | 0.9255 | 0.9603 |
| Cardiotocography | 0.6712 | 0.6353 | 0.6756 | 0.8030 | 0.8001 |
| Census | 0.7371 | 0.7479 | 0.7309 | 0.7247 | 0.7581 |
| Fraud | 0.8739 | 0.9206 | 0.9076 | 0.9006 | 0.9357 |
| Glass | 0.7029 | 0.6686 | 0.7206 | 0.7206 | 0.7225 |
| Ionosphere | 0.9525 | 0.9579 | 0.9626 | 0.9626 | 0.9726 |
| Mammography | 0.8846 | 0.7879 | 0.8558 | 0.8911 | 0.9053 |
| Nslkdd | 0.8603 | 0.8723 | 0.8547 | 0.8661 | 0.8780 |
| Optdigits | 0.9875 | 0.9865 | 0.9786 | 0.9666 | 0.9947 |
| Pendigits | 0.9929 | 0.9943 | 0.9956 | 0.9955 | 0.9919 |
| Pima | 0.7212 | 0.6944 | 0.6909 | 0.7362 | 0.7639 |
| Satellite | 0.7940 | 0.7996 | 0.8021 | 0.7953 | 0.7962 |
| Satimage-2 | 0.9982 | 0.9973 | 0.9975 | 0.9965 | 0.9992 |
| Shuttle | 0.9984 | 0.9982 | 0.9977 | 0.9976 | 0.9975 |
| Thyroid | 0.9795 | 0.9809 | 0.9752 | 0.9817 | 0.9804 |
| Wbc | 0.9755 | 0.9630 | 0.9809 | 0.9808 | 0.9814 |
| Wine | 0.9384 | 0.9538 | 0.9538 | 0.9538 | 0.9538 |
| Average | 0.8745 | 0.8701 | 0.8785 | 0.8916 | 0.9044 |

Table 13: AUC-PR results with different batch size.

| DataSet | batch size=32 | batch size=64 | batch size=128 | batch size=256 | batch size=512 |
|---|---|---|---|---|---|
| Arrhythmia | 0.5826 | 0.5745 | 0.5880 | 0.6100 | 0.6107 |
| Breastw | 0.9877 | 0.9934 | 0.9966 | 0.9952 | 0.9952 |
| Campaign | 0.4994 | 0.4679 | 0.4940 | 0.5176 | 0.6040 |
| Cardio | 0.7073 | 0.7473 | 0.7459 | 0.7960 | 0.8489 |
| Cardiotocography | 0.5962 | 0.5634 | 0.6089 | 0.7189 | 0.6993 |
| Census | 0.2233 | 0.2241 | 0.2153 | 0.1954 | 0.2420 |
| Fraud | 0.2692 | 0.6250 | 0.6121 | 0.5561 | 0.5141 |
| Glass | 0.1652 | 0.1357 | 0.1917 | 0.1917 | 0.1905 |
| Ionosphere | 0.9690 | 0.9717 | 0.9742 | 0.9742 | 0.9802 |
| Mammography | 0.3808 | 0.2395 | 0.3098 | 0.4108 | 0.4755 |
| Nslkdd | 0.8872 | 0.9046 | 0.8712 | 0.8821 | 0.9085 |
| Optdigits | 0.7590 | 0.7211 | 0.6543 | 0.5015 | 0.8885 |
| Pendigits | 0.8017 | 0.8628 | 0.8978 | 0.8906 | 0.8258 |
| Pima | 0.6966 | 0.6861 | 0.6838 | 0.7417 | 0.7389 |
| Satellite | 0.8486 | 0.8509 | 0.8518 | 0.8528 | 0.8532 |
| Satimage-2 | 0.9524 | 0.9504 | 0.9115 | 0.8691 | 0.9850 |
| Shuttle | 0.9705 | 0.9627 | 0.9548 | 0.9500 | 0.9479 |
| Thyroid | 0.8296 | 0.8378 | 0.8367 | 0.8422 | 0.8417 |
| Wbc | 0.8720 | 0.8512 | 0.8850 | 0.8797 | 0.8887 |
| Wine | 0.9208 | 0.9335 | 0.9335 | 0.9335 | 0.9335 |
| Average | 0.6960 | 0.7052 | 0.7108 | 0.7155 | 0.7486 |

Table 14: AUC-ROC results with different designs of diversity loss.

| DataSet | Euclidean Distance | Cosine Similarity | Dot product | Ours w/o Log | Ours |
|---|---|---|---|---|---|
| Arrhythmia | 0.7784 | 0.8114 | 0.7035 | 0.7462 | 0.8114 |
| Breastw | 0.9889 | 0.9952 | 0.9927 | 0.9943 | 0.9955 |
| Campaign | 0.8755 | 0.8191 | 0.6959 | 0.6768 | 0.8902 |
| Cardio | 0.8544 | 0.9391 | 0.9270 | 0.9293 | 0.9603 |
| Cardiotocography | 0.7969 | 0.7987 | 0.8454 | 0.8617 | 0.8001 |
| Census | 0.7269 | 0.7322 | 0.6070 | 0.7104 | 0.7581 |
| Fraud | 0.9270 | 0.9305 | 0.9212 | 0.9228 | 0.9357 |
| Glass | 0.5676 | 0.6657 | 0.7500 | 0.6902 | 0.7225 |
| Ionosphere | 0.9546 | 0.9592 | 0.8374 | 0.8680 | 0.9726 |
| Mammography | 0.8127 | 0.8682 | 0.8424 | 0.8751 | 0.9053 |
| Nslkdd | 0.8075 | 0.8657 | 0.7467 | 0.8557 | 0.8780 |
| Optdigits | 0.9951 | 0.9951 | 0.9929 | 0.9925 | 0.9947 |
| Pendigits | 0.9837 | 0.9701 | 0.8892 | 0.9550 | 0.9919 |
| Pima | 0.7270 | 0.6987 | 0.6963 | 0.6724 | 0.7639 |
| Satellite | 0.7976 | 0.7966 | 0.6816 | 0.6886 | 0.7962 |
| Satimage-2 | 0.9994 | 0.9986 | 0.9976 | 0.9980 | 0.9992 |
| Shuttle | 0.9972 | 0.9974 | 0.9967 | 0.9970 | 0.9975 |
| Thyroid | 0.9353 | 0.9578 | 0.9430 | 0.9298 | 0.9804 |
| Wbc | 0.8784 | 0.9231 | 0.9127 | 0.9285 | 0.9814 |
| Wine | 0.9615 | 0.9461 | 0.9507 | 0.9260 | 0.9538 |
| Average | 0.8683 | 0.8834 | 0.8465 | 0.8609 | **0.9044** |

Table 15: AUC-PR results with different designs of diversity loss.

| DataSet | Euclidean Distance | Cosine Similarity | Dot product | Ours w/o Log | Ours |
|---|---|---|---|---|---|
| Arrhythmia | 0.5924 | 0.6073 | 0.4531 | 0.5629 | 0.6107 |
| Breastw | 0.9857 | 0.9946 | 0.9887 | 0.9944 | 0.9952 |
| Campaign | 0.5727 | 0.5153 | 0.3960 | 0.3766 | 0.6040 |
| Cardio | 0.7159 | 0.7969 | 0.7864 | 0.7828 | 0.8489 |
| Cardiotocography | 0.7038 | 0.6948 | 0.7371 | 0.7501 | 0.6993 |
| Census | 0.2077 | 0.2135 | 0.1568 | 0.1937 | 0.2420 |
| Fraud | 0.5080 | 0.4730 | 0.4052 | 0.4037 | 0.5141 |
| Glass | 0.1992 | 0.1350 | 0.1679 | 0.1671 | 0.1905 |
| Ionosphere | 0.9703 | 0.9725 | 0.8931 | 0.9016 | 0.9802 |
| Mammography | 0.1533 | 0.3454 | 0.2543 | 0.3700 | 0.4755 |
| Nslkdd | 0.8586 | 0.8888 | 0.8465 | 0.8971 | 0.9085 |
| Optdigits | 0.9025 | 0.9042 | 0.8501 | 0.8443 | 0.8885 |
| Pendigits | 0.6178 | 0.4949 | 0.3258 | 0.5061 | 0.8258 |
| Pima | 0.7268 | 0.7042 | 0.6859 | 0.6773 | 0.7389 |
| Satellite | 0.8544 | 0.8534 | 0.7568 | 0.7655 | 0.8532 |
| Satimage-2 | 0.9807 | 0.9773 | 0.9725 | 0.9663 | 0.9850 |
| Shuttle | 0.9406 | 0.9456 | 0.9460 | 0.9345 | 0.9479 |
| Thyroid | 0.7436 | 0.8060 | 0.7100 | 0.7235 | 0.8417 |
| Wbc | 0.6026 | 0.6567 | 0.6611 | 0.6671 | 0.8887 |
| Wine | 0.9294 | 0.9230 | 0.9323 | 0.9181 | 0.9335 |
| Average | 0.6883 | 0.6951 | 0.6463 | 0.6701 | **0.7486** |

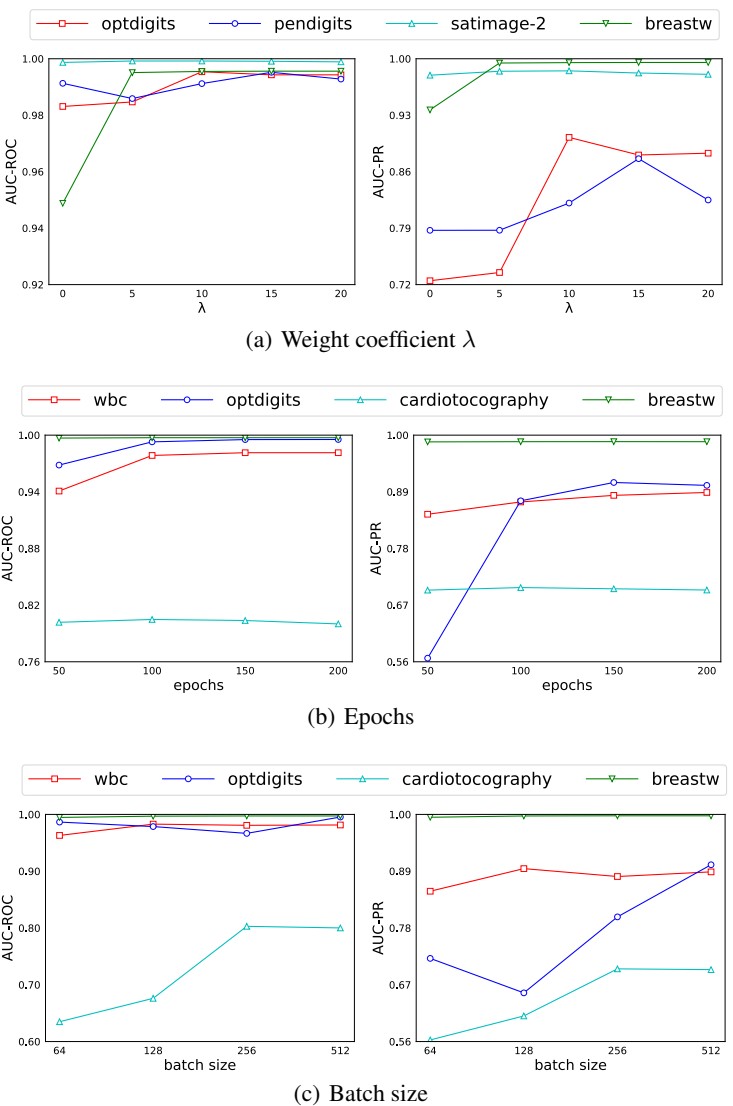

Figure 7: AUC-ROC and AUC-PR performance of parameter sensitivity analysis on the weight coefficient $\lambda$, number of epochs, and batch size.

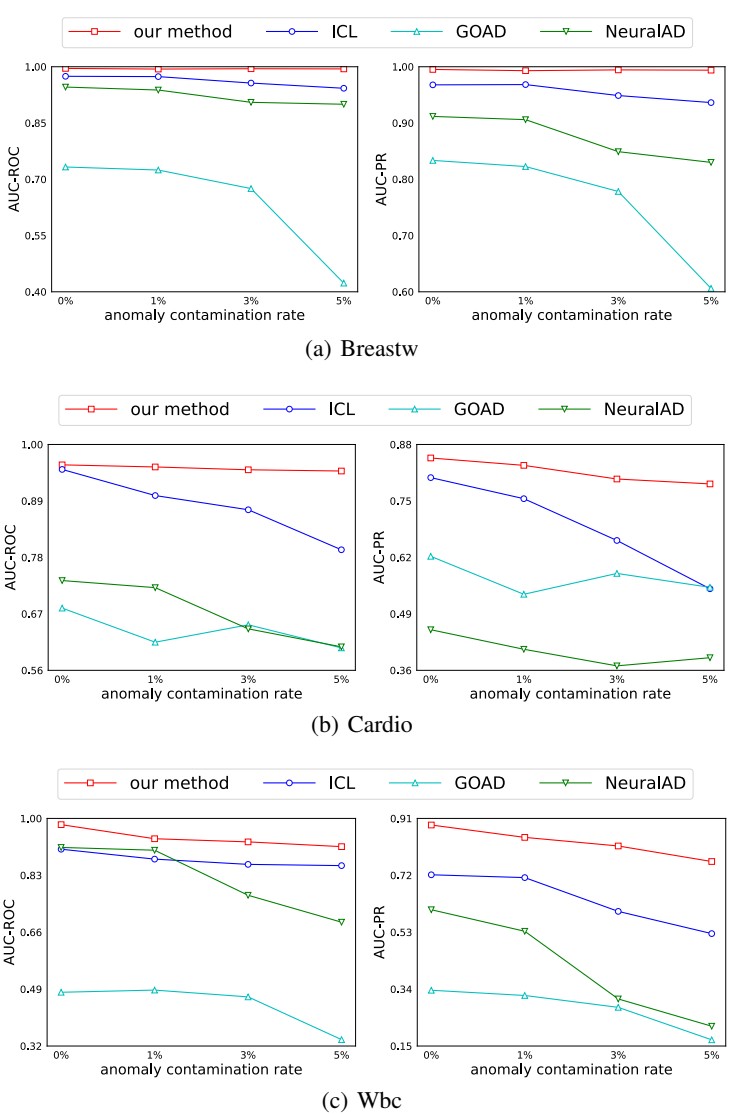

Figure 8: AUC-ROC and AUC-PR performance with different ratio of anomaly contamination.

## C    GENERATION OF FOUR TYPES OF ANOMALIES

Here we present the generation process of four types of anomalies, following (Han et al., 2022).

- **Local anomalies** Using the classic GMM procedure (Milligan, 1985; Steinbuss & Böhm, 2021), we generate normal samples. Then, local anomalies are generated by scaling the covariance matrix by a scaling parameter $\alpha = 5$.
- **Global anomalies** Global anomalies are sampled from a uniform distribution $Unif(\alpha \cdot min(\mathbf{X^k}), \alpha \cdot max(\mathbf{X^k}))$. As is seen from the formula, the boundaries are defined as the $min$ and $max$ of an input feature, e.g., $k$-th feature $X^k$. The hyperparameter $\alpha$ is set to 1.1, which controls the degree of deviation of anomalies.
- **Dependency anomalies** In the generation of dependency anomalies, Vine Copula (Aas et al., 2009) method is applied to model the dependency structure of normal data. The probability density function of generated anomalies is set by removing the modeled dependency, which completes independence(see (Martinez-Guerra & Mata-Machuca, 2016)). Specifically, we use Kernel Density Estimation (KDE) (Hastie et al., 2009) to estimate the probability density function of features and generate normal samples.
- **Clustered anomalies** We scale the mean feature vector of normal samples by $\alpha = 5$, i.e., $\hat{\mu} = \alpha\hat{\mu}$. The hyperparameter $\alpha$ controls the distance between anomaly clusters and the normal. Scaled GMM is employed to generate anomalies.

## D    OTHER DISCUSSIONS

**Latency of training and inference** Table.16 displays the training and inference times for a batch of data of different dimensions, where we utilize generated data ranging from 10 to 10,000 dimensions, batch size is fixed to 64. The experiments were conducted on a single Tesla V100 GPU.

Table 16: Latency of training/inference on different dimensions of data

|  | data dimension | 10 | 50 | 100 | 500 | 1000 | 5000 | 10000 |
|---|---|---|---|---|---|---|---|---|
| Latency [ms] | Training | 8.49 | 8.91 | 9.55 | 17.69 | 29.21 | 224.12 | 785.98 |
|  | Testing | 3.58 | 4.06 | 4.44 | 12.94 | 26.96 | 211.43 | 755.69 |

**Lightweight Decoder** In the CV domain, it has been proven that utilizing a lightweight decoder can boost the performance of MIM He et al. (2022); Xie et al. (2022). We also try changing MCM in this aspect by replacing the decoder with a one-layer head,

In Table.17, we present the detailed results for MCM and the altered structures. As is displayed in the table, this change produces negative effects on most datasets, leading to a drop in average AUC-ROC and AUC-PR. The reasons for this discrepancy are as follows: For image tasks, the encoder extracts high-level semantic information, making it easy to reconstruct lower-level information like pixels, and using a lightweight decoder does not affect the performance. However, in tabular data, cells typically contain various types of information (for example, each column in tabular data has its own practical meaning). Therefore, to reconstruct these finer-grained tabular data, a more powerful decoder is needed to ensure reconstruction capability.

**Pure Masked Prediction** Also, calculating reconstruction error only on masked parts, known as pure masked prediction, is often utilized in classic methods. Following the same setting, we assign a weight $1 - m_j^i$ to the reconstruction error of each feature, where $m_j^i$ is a value from 0 to 1, denoting the degree of mask for $j$th feature of $i$th sample.

The results in Table.17 show that this change also results in poorer performance. Our explanation for this phenomenon is as follows: The unmasked part is actually reconstruction task, while the masked part is for prediction. In methods like MAE designed for images, reconstruction is relatively simple, and adding it can even hinder performance. Focusing solely on the prediction part can indeed compel the model to become stronger. However, tabular data has a more complex structure: there may be relationships between columns, rows, and individual cells, and it's also necessary to learn semantics within each cell. Thus, even reconstructing the unmasked part is challenging because of

these interdependencies. Therefore, improving the reconstruction of the unmasked part by capturing these relationships is also crucial and can enhance the model's performance.

Table 17: AUC-ROC and AUC-PR results on MCM compared to the other two altered structures.

| | Pure masked prediction | | Lightweight decoder | | Ours | |
|---|---|---|---|---|---|---|
| Dataset | AUC-ROC | AUC-PR | AUC-ROC | AUC-PR | AUC-ROC | AUC-PR |
| Arrhythmia | 0.7865 | 0.5683 | 0.7765 | 0.5964 | **0.8114** | **0.6107** |
| Breastw | 0.9953 | 0.9952 | **0.9973** | **0.9973** | 0.9955 | 0.9952 |
| Cardio | 0.8665 | 0.6822 | 0.8968 | 0.7630 | **0.9603** | **0.8489** |
| Census | 0.7285 | 0.1955 | 0.7199 | 0.2073 | 0.7581 | **0.2420** |
| Campaign | 0.8717 | 0.5675 | 0.8010 | 0.5067 | **0.8902** | **0.6040** |
| Cardiotocography | **0.8388** | **0.7282** | 0.7608 | 0.6578 | 0.8001 | 0.6993 |
| Fraud | **0.9412** | 0.4434 | 0.8963 | 0.5363 | 0.9357 | **0.5141** |
| Glass | 0.5971 | 0.1100 | 0.6245 | 0.1199 | **0.7225** | **0.1905** |
| Ionosphere | 0.9681 | 0.9775 | 0.9523 | 0.9695 | **0.9726** | **0.9802** |
| Mammography | 0.8878 | 0.4203 | 0.8895 | 0.4484 | **0.9053** | **0.4755** |
| Nslkdd | 0.8276 | 0.8683 | 0.8699 | 0.8943 | **0.8780** | **0.9085** |
| Optdigits | 0.8776 | 0.1946 | 0.9569 | 0.4081 | **0.9947** | **0.8885** |
| Pima | 0.9890 | 0.7331 | **0.9934** | **0.8766** | 0.7639 | 0.7389 |
| Pendigits | 0.6687 | 0.6858 | 0.7029 | 0.6902 | **0.9919** | **0.8258** |
| Satellite | 0.7675 | 0.8371 | **0.7999** | 0.8528 | 0.7962 | **0.8532** |
| Satimage-2 | 0.9980 | 0.9714 | 0.9968 | 0.9694 | **0.9992** | **0.9850** |
| Shuttle | 0.9971 | 0.9391 | 0.9972 | 0.9425 | **0.9975** | **0.9479** |
| Thyroid | 0.9770 | 0.8356 | 0.8945 | 0.5349 | 0.9804 | **0.8417** |
| Wbc | 0.9622 | 0.8157 | 0.9093 | 0.6679 | **0.9814** | **0.8887** |
| Wine | 0.9584 | 0.8827 | **0.9661** | **0.9394** | 0.9538 | 0.9335 |
| Average | 0.8812 | 0.6881 | 0.8701 | 0.6789 | **0.9044** | **0.7486** |

# E DATASET STATISTICS

The Table.18 shows the numebr of samples, the dimension, and the number of anomalies of each dataset used.

Table 18: Details of datasets used.

| Dataset | Samples | Dims | Anomaly |
|---|---|---|---|
| Arrhythmia | 452 | 279 | 66 |
| Breastw | 683 | 9 | 239 |
| Campaign | 41188 | 62 | 4640 |
| Cardio | 1831 | 21 | 176 |
| Cardiotocography | 2114 | 21 | 466 |
| Census | 299285 | 500 | 18568 |
| Fraud | 284807 | 29 | 492 |
| Glass | 214 | 9 | 9 |
| Ionosphere | 351 | 33 | 126 |
| Mammography | 11183 | 6 | 260 |
| NSL-KDD | 148517 | 122 | 77054 |
| Optdigits | 5216 | 64 | 150 |
| Pendigits | 6870 | 16 | 156 |
| Pima | 768 | 8 | 268 |
| Satellite | 6435 | 36 | 2036 |
| Satimage-2 | 5803 | 36 | 71 |
| Shuttle | 49097 | 9 | 3511 |
| Thyroid | 3772 | 6 | 93 |
| Wbc | 278 | 30 | 21 |
| Wine | 129 | 13 | 10 |

## F    IMPLEMENTATION DETAILS

We train and test the network on two Tesla T4 GPUs on the Ubuntu18.04 system. Our code is implemented based on PyTorch 1.10.2 framework with Python 3.6. Other critical package requirements include torchvision 0.11.3, numpy 1.23.5, pandas 1.5.3, and scipy 1.10.1.

## G    VISUALIZATION OF MASKS GENERATED

Figure.9 shows masks of one normal sample as part of the case study presented in Section 5, **correlations between features** section.

Figure.10 shows the comparison of masks with and without the inclusion of diversity loss. The visual analysis reveals a notable increase in mask diversity when diversity loss is included. In contrast, masks without diversity loss exhibit a concentration of values around 0.5, indicating lower distinctiveness. Using diversity loss helps capture the diverse correlations present within normal data, thereby contributing to the detection of anomalies.

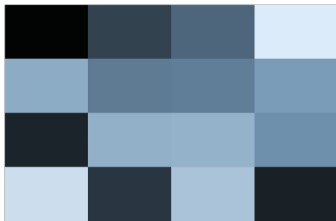
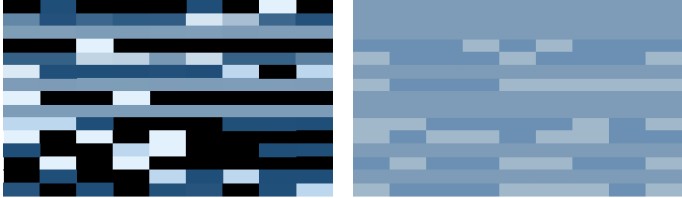

Figure 9: Visualization of masks on one normal sample. The darker/lighter color indicates a value near 0/1.

Figure 10: Multiple masks on one sample of Breastw dataset when training with (left) and without (right) diversity loss. A darker/lighter color means a value closer to 0/1, each row represents a different kind of mask.

## H    DISCUSSION OF DEGENERATION LEARNING

Firstly, we define the mask degeneration problem as follows: for a mask vector generated for a sample, all the values are 1, then nothing is masked, indicating the occurrence of mask degeneration. Furthermore, we can extend the degeneration problem to the cases where the values in a mask vector are very close or even identical. This is because in such situations, the effect of the mask on the input is more like a uniform scaling, rather than selectively masking some features and reconstructing them with other features. Therefore, this phenomenon can also be considered as the occurrence of mask degeneration.

We conduct both qualitative and quantitative experiments on this problem. See Figure.9 for qualitative results, it is observed that when diversity loss is not applied (i.e., the right figure), more than half of the mask vectors show degeneration issues, where the all mask values are identical. When training with diversity loss (left figure), none of the mask vectors exhibit this issue, and there is significant diversity among the masks. For quantitative results, we use the KL divergence between a mask vector and the uniform distribution to define the degree of degeneration:

$$D_{KL}(\mathbf{u}||\mathbf{m}) = -\frac{1}{F}\sum_{i=1}^{F} log(m_i) - H(\mathbf{u}) \tag{6}$$

where $\mathbf{u}$ indicates the uniform distribution, $\mathbf{m}$ is a mask vector with length F. The intuitive interpretation of this expression is that as the values of the mask vector become closer, their KL divergence with the uniform distribution will be smaller, indicating more severe degeneration. Taking a sample from the Breastw dataset as an example, when trained with and without diversity loss, the degeneration degree is 9.0724e-06 and 0.8730. This suggests that under the constraint of diversity loss, MCM can effectively avoid mask degeneration problem.

