# OpenReview forum: "MCM: Masked Cell Modeling for Anomaly Detection in Tabular Data"
_ICLR.cc/2024/Conference — ICLR 2024 poster_

### Official Review · Reviewer_8h9M · 2023-10-15

**Soundness:** 3 good
**Presentation:** 2 fair
**Contribution:** 3 good
**Rating:** 6
**Confidence:** 3

**Summary:**

This paper adapts masked cell modelling for anomaly detection in tabular data, with learnable and ensembled masks for further performance improvement.

**Strengths:**

MLM for anomaly detection is relatively under-explored and for tabular data it is a novel avenue for research.

The results are promising and comprehensive, including the ablation study.

**Weaknesses:**

Some aspects of the methodology and decisions are not entirely clear. Please see questions for further details.

**Questions:**

What is the importance of using soft masks instead of binary masks? It appears that data transformation through soft masking could cause the sample to break from the normal patterns and actually be an anomaly itself, but be treated like a normal sample in the training process. Could this hinder model training?

Why is learnable masking so important for AD, when random masks are fine for other tasks?

The batch size is very large, and is an important parameter as it affects the dimensionality of M. How does the methodology perform with a smaller batch size? Also, can the methodology still work if the batch size changes between training and test sets?

---

> ### Author Response · Authors · 2023-11-17
> **Response to Reviewer 8h9M**
>
> We thank you for your reviews and address your concerns as follows:
>
> **Q1**: What is the importance of using soft masks instead of binary masks? It appears that data transformation through soft masking could cause the sample to break from the normal patterns and actually be an anomaly itself, but be treated like a normal sample in the training process. Could this hinder model training?
>
> **A1**: It worth noting that the anomaly score of one sample is the average reconstruction error of its each masked version. So the masked data itself is not important, it can be of any pattern, we only focus on whether the masked data can be reconstructed well. Similarly, the model is trained to generate masks under which the masked data can be reconstructed well for normal data. Therefore, under generated masks, normal data can be reconstructed better than anomalies, such that anomalies can be detected. To sum up, the pattern of masked data will not hinder model training. When applying soft masks, the model can not only choose which features to mask, but also the degree of masking. This brings more flexibility for the model and is conductive to learning diverse and optimal masks under which the masked normal data can be reconstructed better than anomalies.
>
>
>
> **Q2**: Why is learnable masking so important for AD, when random masks are fine for other tasks?
>
> **A2**: This is mainly due to the difference in **data type** and  **the purpose of masking**. In CV domain, a simple random masking strategy with a high masking ratio works quite well [1]. Because all the features of image data are pixels, which are **homogeneous and highly relevant**, no matter where to mask, predicting a large proportion of masked patches with just a small proportion of unmasked patches is meaningful for the model to **understand an image**, such that useful representations for downstream tasks can be learnt. On the contrary, the features of tabular data are **heterogeneous**, each feature has a different meaning, some of them are totally **irrelevant**. Our purpose is to **capture intrinstic correlations** in normal data by finding using which unmasked features can reconstruct the left masked features well. Random masking strategy is easy to produce meaningless masks for this purpose. But by learnable masking strategy, the mask generator can generate optimal masks for our purpose. Thus, correlations that can characterize normal data will be captured and anomalies can by judged by whether deviating from such correlations.
>
> [1] Masked autoencoders are scalable vision learners. CVPR 2022
>
>
>
> **Q3**: The batch size is very large, and is an important parameter as it affects the dimensionality of M. How does the methodology perform with a smaller batch size? Also, can the methodology still work if the batch size changes between training and test sets?
>
> **A3**: In our method, there are only two matrix operations involving M, the first is element-wise product between M and the original data to produce masked data, the second is the calculation of diversity loss. Both of them are calculated independently for each sample and do not involve calculations between different data. Therefore, although batch size affects the dimensionality of M, it does not affect the overall time and complexity of matrix operations. So a large batch size is quite acceptable, especially considering that the dimensions of tabular data are mostly relatively low. However, we still supplement the parameter sensitivity experiments for batch size in **Table 12** and **Table 13** in the latest version. As is shown, the average AUC-PR on the 20 benchmark datasets with batch size of **{64, 128, 256, 512}** are **{0.7052, 0.7108, 0.7155, 0.7486}**. The performance does decrease as the batch size decreases, but even with a batch size of 64, our method can still surpass all  the compared methods and achieve SOTA performance. Moreover, as we illustrated above, our method is calculated for each data separately, so the batch size of the test set can be any value. Of course, it can also be different from the batch size of the training set, and it will not affect the final results.

---

### Official Review · Reviewer_yu8Y · 2023-10-27

**Soundness:** 4 excellent
**Presentation:** 3 good
**Contribution:** 3 good
**Rating:** 8
**Confidence:** 4

**Summary:**

The manuscript proposes a new method for anomaly detection in tabular data. The proposed method detects anomalies based on the reconstruction error of appropriately trained autoencoder. In the proposed training procedure, a tabular input is first masked according to diverse masking strategies to produce a batch of masked inputs. Diverse masks are produced by a set of models trained to output masks of high variety. The obtained masked input is then fed to an autoencoder which reconstructs the initial input. The corresponding optimization objective consists of a reconstruction loss and a loss which enforces the diversity between the generated masks. During inference, inputs are marked as anomalies if they are unsuccessfully reconstructed which is indicated by the high reconstruction loss. The proposed method achieves strong experimental results under considered setups which include real and synthetic test anomalies.

The manuscript claims the following contributions:

C1. Masked modeling framework for anomaly detection in tabular data.

C2. Masking strategy which captures underlying high-level correlations in the normal data.

C3. Strong experimental results on tabular data.

**Strengths:**

S1. The manuscript is well-written and easy to follow.

S2. Masked modeling for tabular data makes sense.

S3. The proposed data-dependent masking strategy outperforms other more naive masking strategies.

S4. The presented empirical results are indeed strong and ablations are extensive.

**Weaknesses:**

W1. Diversity loss (Eq. 5) has log-sum-exp form. The motivation for this design choice is not clear since other formulations may also be effective. Also, ablating different designs of diversity loss are missing.

**Questions:**

Q1. Are there any limitations of the proposed method?

---

> ### Author Response · Authors · 2023-11-17
> **Response to Reviewer yu8Y**
>
> We appreciate the kind and constructive comments and answer your question as follows:
>
> **Q1** Are there any limitations of the proposed method?
>
> **A1**: In the current field of deep learning, most attention is paid to general large models that can be trained on various datasets and applied to downstream tasks. Although our network architecture and parameters almost do not need to change for different datasets, our method still requires training a model from scratch for each dataset, which can not combine knowledge from different datasets. In the future, we will try to change our network to a Transformer-like architecture, train one general model to learn knowledge from different datasets, and address various tabular AD tasks of different dimensions and areas by just using one model.

---

> > ### Comment · Reviewer_yu8Y · 2023-11-22
> > **Weaknesses not addressed?**
> >
> > The authors still did not respond to (W1.).

---

> > > ### Author Response · Authors · 2023-11-22
> > > **Response to Reviewer yu8Y about Weakness**
> > >
> > > **W1**: Diversity loss (Eq. 5) has log-sum-exp form. The motivation for this design choice is not clear since other formulations may also be effective. Also, ablating different designs of diversity loss are missing.
> > >
> > > **A1**: **First**, the motivation of our log-sum-exp formulation is derived from the InfoNCE loss of contrastive learning [1]. In the expression of InfoNCE loss, the numerator aims to increase the similarity between positive pairs, while the denominator aims to decrease the similarity between negative pairs. As we desire all masks of a sample to be dissimilar, all pairs of masks can be considered as negative pairs. Thus, we adopted the form of the denominator in InfoNCE, which is in log-sum-exp form. Experiments show the design leads to satisfying performance.
> > >
> > > **Second**, to investigate different designs of diversity loss, we conduct ablation experiments on 20 benchmark datasets. Because diversity loss is used to decrease the similarity of different masks of a sample, we set the following formulations to calculate the similarity of different masks  : *1. Euclidean distance 2. Cosine similarity 3. Dot product 4. Our proposed form without log operation 5. Ours*.  As shown in **Table 14** and **Table 15** in Appendix of the latest version, the average AUC-ROC are **0.8683, 0.8834, 0.8465, 0.8609, 0.9044**, while the average AUC-PR are **0.6883, 0.6951, 0.6463, 0.6701, 0.7486,** respectively, demonstrating the effectiveness of our design.
> > >
> > > [1] Momentum Contrast for Unsupervised Visual Representation Learning. CVPR 2020

---

### Official Review · Reviewer_cK19 · 2023-10-31

**Soundness:** 3 good
**Presentation:** 3 good
**Contribution:** 3 good
**Rating:** 6
**Confidence:** 4

**Summary:**

The paper proposes a new method for improving anomaly detection via self-supervised learning with masking modeling for tabular data. The key idea is to train a network on the clean input to generate the mask, and then apply the mask into these inputs before feeding them through the auto-encoder to reconstruct the clean inputs. The mask matrices are designed as soft masks via sigmoid function. To avoid the collapse of the generator in generating trivial and redundant mask matrices, the authors regularize the reconstruction loss with diversity loss which penalizes the masking matrices to be far away. The experiments are conducted with 20 benchmark datasets and compared with 9 baselines. The experimental results are encouraging compared to baselines on benchmark datasets. The paper also shows a number of ablation studies to understand the impacts of each individual proposed component and of masking strategy, and how it detects different types of anomalies. In addition, the authors also perform the analysis on robustness of number of masking matrices and contamination and some other studies relating to the feature correlation.

**Strengths:**

The proposed method is sensible and novel to me and the results are quite encouraging. The experiments and ablation study are quite extensive. Potentially, this idea of learning the masking matrices might be applicable to other domains although there might be some difficulties regarding computation when dealing with higher-dimensional data.

**Weaknesses:**

In Table 2, the ensemble appears not to contribute a lot to performance. Would it be beneficial to have another option similar to Task E but Ensemble is turned off?

Table 3, have the authors studied the impact of parameter $\lambda$ on the overall performance?

It would be beneficial to show the mask of normal data as well would be beneficial in Fig. 5, and also the variety of masks generated by the mask generator?

Would it be beneficial to investigate the model parameters and latency of training/inference of compared methods? The diversity loss is useful but the computation cost is the concern as it has to deal with matrices. Has the author studied the scalability of the proposed method in terms of data dimension, e.g., how the proposed method handles with tabular data scaled at very high dimension?

What does x-axis represent in Fig. 6?

**Questions:**

See above.

---

> ### Author Response · Authors · 2023-11-17
> **Response to Reviewer cK19 (Part 1)**
>
> We thank you for your reviews and address your concerns as follows:
>
>  **Q1**: In Table 2, the ensemble appears not to contribute a lot to performance. Would it be beneficial to have another option similar to Task E but Ensemble is turned off?
>
>  **A1**: **First**, diversity loss is used to encourage the diversity of different masks, when ensemble is turned off, diversity loss is useless because there is only one mask. Therefore, this option is the same as Task C. **Second**, compared to Task C, Task D applies ensemble learning but contributes little, which suggests that simply increasing the number of masks doesn’t work, as the model tends to generate redundant masks. **Third**, compared to Task C, Task E applies ensemble learning with diversity loss and achieves a remarkable performance boost. This illustrates the significance of the interaction between ensemble and diversity, which is necessary to capture diverse correlations that can characterize normal data better.
>
>
>
> **Q2**: Table 3, have the authors studied the impact of parameter λ on the overall performance?
>
> **A2**:  In the original version, we have studied the impact of λ on several datasets, but have not studied the overall performance. We agree with you and add the experiments with λ taking **1, 5, 10, 20, 50** and **100** on all the datasets to study the impact of λ on the overall performance. The average AUCPR are **0.6928, 0.7105, 0.7135, 0.7255, 0.7089** and **0.7062**, respectively. The detailed results of each dataset are provided in **Table 10** and **Table 11** of the Appendix in the new version. These results show that our method is not sensitive to λ, a fixed value of 20 can achieve superior performance and outperform other masking strategies in Table 3 by a large margin.
>
> Moreover, in comparison to that λ is adjusted individually for each dataset, the average AUCPR of a fixed λ only decreased by 0.023, which still surpasses all baselines. Considering the only two parameters tuned for different datasets in our method are learning rate and λ, above results further prove that our method is very insensitive to parameters.
>
>
>
> **Q3**: It would be beneficial to show the mask of normal data as well in Fig. 5, and also the variety of masks generated by the mask generator?
>
> **A3**: Thank you for your suggestion. We add the masks of one normal sample in **Fig.9**. Moreover, to show the variety of masks generated by the mask generator, we select one sample in Breastw dataset and draw the 15 masks generated when training with and without diversity loss. As shown in **Fig.10**, without diversity loss, more than half masks are identical and redundant, other masks are also relatively similar to each other. Differently, when applying diversity loss, the diversity of generated masks is significantly increased.

---

> ### Author Response · Authors · 2023-11-17
> **Response to Reviewer cK19 (Part 2)**
>
> **Q4**: Would it be beneficial to investigate the model parameters and latency of training/inference of compared methods? The diversity loss is useful but the computation cost is the concern as it has to deal with matrices. Has the author studied the scalability of the proposed method in terms of data dimension, e.g., how the proposed method handles with tabular data scaled at very high dimension?
>
> **A4**: **First**,  we conduct experiments on optdigits dataset which contains 5216 samples of 64 dimensions, and compare our method with two latest self-supervised learning based methods NeutralAD [1] and ICL [2]. The parameters of NeutralAD, ICL, and our proposed MCM are **0.30M, 0.02M, and 0.41M**. The time required to train one epoch is **0.01s, 3.51s, and 3.76s**, while the inference times are **0.79s, 0.28s, and 0.20s**, respectively (on one Tesla V100 GPU). As is shown, these three models have their own advantages, our method performs better in inference time, which is more critical in practical applications. It is also worth mentioning that in the current large models era, compared to mainstream models in the fields of NLP and CV (such as 340M BERT-L and 307M ViT-L), the parameters and training/testing time of our model are several orders of magnitude smaller, and the differences between methods in our area are quite marginal.
>
> **Second**, we use $n$, $t$, $d$ to denote batch size, the number of masks, and data dimension. Although diversity loss has to deal with matrices, it calculates the similarity between masks of each independent data and does not involve calculations between different data. The complexity of one similarity calculation is $O(d)$. For each data, $t(t-1)/2$ times will be calculated. So the computational complexity of diversity loss is $O(n$$t^2$$d)$, which is linear with the data dimension $d$ and the cost is affordable with the increasing of $d$.
>
> **Third**, we generate some data with dimensions ranging from 10 to 10,000 to evaluate the scalability of the proposed method and add the results in **Table 16**. This table reveals two important points: (i) For the dimensions in the range of 10-1000, where the vast majority of tabular data belongs to, the growth of time with dimensions is lower than the linear growth rate. (ii) When dealing with data of 10,000 dimensions, although this dimension has far exceeded the general tabular data, the training and inference times for a batch are less than 1s (on one Tesla V100), which is quite acceptable. The reason is that except for the input and output layers, our network architecture does not change for data of different dimensions. Therefore, the time only increases in the first and the last layers as the dimension increases.
>
> [1] Neural transformation learning for deep anomaly detection beyond images. ICML 2021
>
> [2] Anomaly detection for tabular data with internal contrastive learning. ICLR 2022
>
>
> **Q5**: What does the x-axis represent in Fig. 6?
>
> **A5**: In **Fig.6**, the x-axis represents the value of the feature, and we have added this information to Fig.6 in the new version. Thanks for the heads up.

---

### Meta-Review · Area_Chair_RdLt · 2023-12-07

**Metareview:**

This paper proposes a method for anomaly detection in tabular data through masking modeling. In order to enhance the performance of anomaly detection by better capturing the correlation between features, the authors train a soft mask generator to create multiple masks. These masks are then used to train a masked auto-encoder, capable of effectively reconstructing each mask on average in an end-to-end fashion.

Three reviewers have thoroughly reviewed this paper, unanimously agreeing that it is easy to follow, innovative, and yields quite encouraging results. Consequently, they recommend acceptance. While the application of the masked autoencoder for anomaly detection may be considered a marginal contribution, the paper demonstrates significant contributions in achieving excellent performance on tabular data and showcasing the effectiveness of masked modeling through soft masking.

However, for further improvement in the paper's quality in its final version, I strongly suggest to include the following:

1. For the completeness of the papers, explicitly specify in the main text, rather than in captions, how the finally learned masked modeling is utilized for anomaly detection.
2. Experimentally demonstrate that the use of soft masks, as opposed to binary masks trained using methods like Gumbel-Softmax, is more effective for the purpose of anomaly detection.
3. Considering the end-to-end learning scenario for the mask generator, there is concern that training in the direction of minimizing the reconstruction loss might lead to degenerate learning, where nothing is masked. Although diversity loss helps mitigate this to some extent, including discussions and specific experimental results for situations where such issues do not arise explicitly would be beneficial.

**Justification For Why Not Higher Score:**

When considering the core ideas of this paper, I find them somewhat trivial, leaning towards the borderline and closer to a slight rejection. However, given the unanimous opinions of the reviewers, my stance is not strong enough to overturn them. Therefore, I have tentatively proposed acceptance. Instead, I have included my concerns and considerations regarding my viewpoint in the last section of the meta-review.

**Justification For Why Not Lower Score:**

review scores are good enough (and moreover, all reviewers are positive)

---

### Decision · Program_Chairs · 2024-01-16

Accept (poster)